# Impact of COVID-19 on mucormycosis presentation and laboratory values: A comparative analysis

Sepideh Hejazi[1], Ali Gholampour Kargar[2], Sahar Ravanshad[3], Arash Ziaee[4], Maryam Emadzadeh[5], Mona Kabiri[6], Reza Khoshbakht[7], Mohammad Hossein Ahmadi[8], Masoumeh Hosseinpoor[9], Hamed Khosravi[10], Imtiaz Ahmed[10], Mehdi Bakhshaee[11]*

**1** Lung Diseases Research Center, Mashhad University of Medical Sciences, Mashhad, Iran, **2** Student Research Committee, Mashhad University of Medical Sciences, Mashhad, Iran, **3** Department of Internal Medicine, Faculty of Medicine, Mashhad University of Medical Science, Mashhad, Iran, **4** Student Research Committee, Mashhad University of Medical Sciences, Mashhad, IranIran, **5** Department of Community Medicine, Faculty of Medicine, Mashhad University of Medical Sciences, Mashhad, Iran, **6** Nanotechnology Research Center, Pharmaceutical Technology Institute, Mashhad University of Medical Sciences, Mashhad, Iran, **7** Department of Laboratory Sciences, School of Paramedical Sciences, Mashhad University of Medical Sciences, Mashhad, Iran, **8** Department of Laboratory Sciences, Faculty of Paramedical and Rehabilitation Sciences, Mashhad University of Medical Sciences, Mashhad, **9** Sinus and Surgical Endoscopic Research Center, Mashhad University of Medical Sciences, Mashhad, Iran, **10** Department of Industrial & Management Systems Engineering, West Virginia University, Morgantown, **11** Sinus and Surgical Endoscopic Research Center, Mashhad University of Medical Sciences, Mashhad, Iran

* bakhshaeem@mums.ac.ir

## Abstract

### Background

The COVID-19 pandemic has led to an alarming increase in mucormycosis coinfections and its rapid progression. The overlapping risk factors and symptoms between COVID-19 and mucormycosis further complicate prompt detection, which is crucial for patient survival. This study aims to investigate potential differences in mucormycosis progression, initial symptom presentation, and laboratory value alterations in mucormycosis patients with COVID-19 history to enhance diagnostic accuracy and improve outcomes in this complex clinical scenario.

### Methodology

This retrospective cohort study, conducted from April 1, 2021, to March 31, 2022, examined 102 patients diagnosed with mucormycosis at two primary teaching hospitals. Patients were categorized into two groups based on COVID-19 history. Variables included demographic information, clinical parameters, laboratory results, and outcomes. The study compared patient laboratory studies and presentation symptoms between COVID-19 history-positive and COVID-19 history-negative groups, with a particular focus on mortality rates and associated comorbidities such as diabetes, cancer and immunosuppressive treatment.

**Data availability statement:** We have successfully deposited the minimal anonymized dataset necessary to replicate our study findings to Zenodo. The DOI for the public dataset is: https://doi.org/10.5281/zenodo.15115574.

**Funding:** The author(s) received no specific funding for this work.

**Competing interests:** The authors have declared that no competing interests exist.

## Results

Initial clinical presentations differed significantly, eneralized Estimating Equations (GEE) analysis, adjusted for comorbidities, revealed COVID-19 history was associated with increased platelet counts $(P = 0.0311)$ and decreased facial swelling $(P = 0.049)$ and fever symptom reporting $(P < 0.001)$. Cancer history, diabetes, and immunosuppressive treatment also showed significant associations with various clinical and laboratory parameters. Laboratory analysis revealed significant differences between mucormycosis patients with and without COVID-19 history. The COVID-19 history-positive group showed lower WBC counts $(P = 0.002)$, and higher hemoglobin levels $(P < 0.001)$ compared to controls. Diabetes was more prevalent in COVID-19 history-positive patients, while cancer history was more common in controls.

## Conclusion

This study reveals intricate relationships between COVID-19 history, mucormycosis, patient presentation, challenging earlier findings. Mucormycosis patients with COVID-19 history exhibited higher platelet counts and altered symptom presentation. The research highlights varied symptom patterns across patient subgroups and underscores the complexity of interactions between COVID-19, cancer, and diabetes in mucormycosis cases. These findings advocate multivariate analytical approaches to better understand these multifaceted relationships.

## Introduction

Beyond straining global health systems, the COVID-19 pandemic has precipitated severe complications, including a notable surge in mucormycosis cases among infected individuals [1]. Mucormycosis is a rare but potentially fatal fungal infection caused by fungi of the order *Mucorales*, primarily affecting immunocompromised patients. The pandemic inadvertently created conditions favorable for this opportunistic infection: the widespread use of corticosteroids and immunosuppressive therapies in treating COVID-19 patients led to immunosuppression, while the virus itself caused endothelial damage, both of which promoted the proliferation of *Mucorales* fungi [2]. These risks were especially pronounced in patients with preexisting conditions such as diabetes mellitus [3,4]. Consequently, the global incidence of COVID-19-associated mucormycosis (CAM) surged during the second wave of the pandemic, notably in India and Iran, where environmental and regional factors further exacerbated infection rates [2,5].

Multiple factors contribute to the heightened risk of mucormycosis in COVID-19 patients. SARS-CoV-2 infection fosters an environment ideal for Mucorales spore germination due to conditions like hypoxia, metabolic acidosis, elevated ferritin levels, and impaired phagocytic activity of white blood cells. Hyperglycemia—stemming from preexisting diabetes, new-onset conditions, or steroid-induced effects—further exacerbates this risk. Immunosuppression, whether mediated by the virus,

corticosteroid treatment, or underlying diseases, diminishes the body's defenses against fungal infections. Additionally, prolonged hospital stays, and mechanical ventilation increase exposure to fungal spores, compounding the likelihood of infection [6,7].

Diagnosing mucormycosis amid the COVID-19 pandemic poses significant challenges, particularly in low- and middle-income countries where advanced diagnostic tools and specialists are scarce [8–10]. The overlapping symptoms and risk factors of both diseases often lead to underdiagnosis or misdiagnosis. Considering that a mere 12-hour delay in identifying mucormycosis can be fatal [10], timely detection is imperative. The lack of clinical suspicion and difficulties in isolating the causative fungi further hinder prompt diagnosis [3]. Therefore, a deeper understanding of the distinct symptomatology and laboratory patterns of mucormycosis in COVID-19 patients is essential to improve clinical outcomes and guide healthcare providers in resource-limited settings [11]. Mortality rates for CAM remain high, with studies showing that delayed diagnosis and severe co-morbid conditions, such as pneumonia, increase fatal outcomes [12]. Hence, a rapid and reliable diagnostic approach is imperative to improve survival in affected populations [13].

This study aims to investigate the differences in progression and initial symptom presentation of mucormycosis among patients with and without COVID-19. Additionally, we will examine alterations in laboratory values in mucormycosis patients co-infected with COVID-19 compared to those without the COVID-19 virus. While previous research has established the increased risk and prevalence of mucormycosis in COVID-19 patients, often attributing these to factors like corticosteroid use and underlying conditions such as diabetes, our study contributes distinctively by employing multivariate analysis to dissect the complex interplay between COVID-19 history, comorbidities, and the manifestations of mucormycosis. Specifically, we look into how a history of COVID-19 may alter the typical clinical and laboratory presentation of mucormycosis, independent of other known risk factors. Moreover, we bolster the diagnostic implications of these altered laboratory findings by demonstrating their potential to serve as early indicators of mucormycosis in the context of COVID-19, thereby facilitating timely intervention and potentially improving patient outcomes. By addressing these objectives, we hope to contribute valuable insights into the clinical management of mucormycosis in the context of the COVID-19 virus, ultimately aiding in the development of more effective diagnostic and therapeutic approaches.

## Materials and methods

### Study design and setting

This retrospective study received written ethical approval from the Institutional Ethics Committee on December 17, 2021, under protocol number 4001291, and registered with the IRCT code IR.MUMS.MEDICAL.REC.1401.054. This retrospective cohort study analyzed patient data collected during the COVID-19 pandemic. The researchers accessed the data in April 2022, covering the period from April 1, 2021, to March 31, 2022. The data was gathered from two major teaching and research hospitals in eastern Iran: Ghaem Hospital and Imam Reza Hospital. These institutions serve as the main referral medical centers for the eastern region. The study utilized comprehensive patient data extracted from each center's electronic health records (EHR) systems.

### Participants

The study involved 102 patients diagnosed with mucormycosis, confirmed by histopathological examination. Patients were categorized into two groups based on their COVID-19 status: those with a history of COVID-19 (exposure group) and those without (control group). This classification was essential for comparing outcomes across patient demographics and medical histories.

**Inclusion criteria.** Patients included in the study had a definite infection with mucormycosis, confirmed by medical diagnosis laboratories through histopathology, which led to their hospitalization. Patients were classified into the exposure group if they had either a current or historical COVID-19 infection and subsequently developed mucormycosis. A current

COVID-19 infection was defined by a positive PCR test or chest X-ray findings highly suggestive of COVID-19, such as atypical or organizing pneumonia, characterized by patchy or diffuse airspace opacities, including consolidation or ground-glass opacity [14,15]. Additionally, individuals with a history of a positive COVID-19 PCR test or chest X-ray findings indicative of a past COVID-19 infection, who later presented with mucormycosis, were also included. Patients who developed mucormycosis without a positive PCR test and absence of COVID-19 clinical symptoms, such as fever, cough with sputum production, smell and taste disturbances, fatigue, and shortness of breath [16], were placed in the control group. The diagnosis of mucormycosis was confirmed by histopathological examination, revealing broad, pauciseptate, ribbon-like hyphae (5–15 microns in diameter) with irregular branching patterns forming 90-degree angles, matching established diagnostic criteria [17]. COVID-19 molecular detection was performed using the Rotor-Gene Q (Qiagen, Germany) PCR system with the Magcore® (RBC Bioscience Corp., Taiwan) extraction device for sample processing and analysis.

**Exclusion criteria.** Patients were excluded from the study if their information was non-available or incomplete. This ensured that the analysis was based on complete and accurate data, thus maintaining the integrity of the study results.

## Data collection and variables

COVID-19 PCR details, laboratory data, demographic information, plain X-Ray Imaging details, and the presence of COVID-19 clinical symptoms and mucormycosis clinical symptoms were extracted from patients' health records and documented using a structured, patient-specific checklist. Table 1 summarizes the key variables and their characteristics.

## Statistical analysis

Data were analyzed using SPSS software, version 22, and R statistical software through Google Collab. Descriptive statistics such as means, standard deviations, medians, and interquartile ranges were used to describe the quantitative data. For comparisons between the two groups, the Unpaired t-test [18] or its non-parametric equivalent, the Mann-Whitney

**Table 1. Summary of Variables Collected for the Study on Mucormycosis in Post-COVID-19 Patients.**

| Variable | Role | Type | Scale | Definition | Unit |
|---|---|---|---|---|---|
| **Age** | Covariates | Quantitative | Ratio | Recorded based on patient history | Years |
| **Sex** | Covariates | Qualitative | Nominal | Recorded based on patient history | Male/Female |
| **Vital Status** | Covariates | Qualitative | Nominal | Status at the end of the observation period | Deceased/Alive |
| **Diabetes History** | Dependent | Qualitative | Nominal | Based on patient history | Present/Absent |
| **Cancer History** | Dependent | Qualitative | Nominal | Based on patient history | Present/Absent |
| **Immunosuppressive Drug History** | Dependent | Qualitative | Nominal | Based on patient history | Present/Absent |
| **Initial Clinical Complaint of Mucormycosis** | Independent | Qualitative | Nominal | Recorded based on initial symptoms | Present/Absent |
| **White Blood Cell Count (WBC)** | Independent | Quantitative | Ratio | Measured during initial laboratory tests | *1000/µL |
| **Polymorphonuclear leukocytes (PMN)** | Independent | Quantitative | Ratio | Measured during initial laboratory tests | % |
| **Lymphocytes (Lymph)** | Independent | Quantitative | Ratio | Measured during initial laboratory tests | % |
| **Hemoglobin (Hb)** | Independent | Quantitative | Ratio | Measured during initial laboratory tests | g/dL |
| **Platelets (Plt)** | Independent | Quantitative | Ratio | Measured during initial laboratory tests | *1000/µL |
| **Aspartate Aminotransferase (AST)** | Independent | Quantitative | Ratio | Measured during initial laboratory tests | U/L |
| **Alanine Aminotransferase (ALT)** | Independent | Quantitative | Ratio | Measured during initial laboratory tests | U/L |
| **Alkaline Phosphatase (ALP)** | Independent | Quantitative | Ratio | Measured during initial laboratory tests | U/L |
| **Estimated Sedimentation Rate (ESR)** | Independent | Quantitative | Ratio | Measured during initial laboratory tests | mm/h |
| **C-reactive Protein (CRP)** | Independent | Quantitative | Ratio | Measured during initial laboratory tests | mg/L |
| **Lactate Dehydrogenase (LDH)** | Independent | Quantitative | Ratio | Measured during initial laboratory tests | U/L |

test [19], was utilized depending on the data distribution. Chi-square or Fisher's exact test [20] was employed for categorical variables. In addition to these methods, Generalized Estimating Equations (GEE) [21] were implemented to analyze population-averaged effects. These approaches are particularly useful in studies like ours, where interest lies in understanding the effects at the population level rather than at the individual level. To ensure statistical rigor, a significance level of less than 0.05 was maintained throughout the analyses.

## Ethical considerations

The study was conducted in compliance with medical ethics principles and approved by the Ethics Committee of Mashhad University of Medical Sciences. Patient information was managed confidentially, analyzed anonymously. The study was conducted and written in accordance with the STROBE checklist [22](S1 File). The study was approved with the ethics code IR.MUMS.MEDICAL.REC.1401.054. The requirement for patients written consent was waived by the Ethics Committee of Mashhad University of Medical Sciences and verbal informed consent was obtained from all participants prior to their involvement in the study.

## Results

### Overview of patients' demographics and clinical characteristics

Table 2 provides an overview of patient demographics, highlighting differences based on COVID-19 status. It details the even distribution of gender across the study and significant age differences between the groups, which could potentially influence clinical outcomes.

We examined laboratory parameters to understand the biochemical and hematological impacts of COVID-19 on mucormycosis patients. Table 3 presents a comparative analysis of various laboratory values between the mucormycosis patients with positive COVID-19 history and those without COVID-19 history. These laboratory results offer insights into the physiological differences between the groups, which may influence the clinical outcomes of mucormycosis infection.

Significant differences were observed in the laboratory values between the COVID-19 history-positive and non-COVID groups. The COVID-19 history-positive group exhibited lower white blood cell counts (P = 0.002) and significantly higher hemoglobin levels (P < 0.001) compared to the control group. However, other laboratory measures did not show significant differences, such as lymphocyte percentages, ESR, CRP, LDH, AST, ALT, and ALP.

### Mortality analysis and past medical history

In examining the impact of COVID-19 on the progression and outcomes of mucormycosis, we analyzed mortality rates across the two groups. The COVID-19 history-positive group displayed a higher mortality rate (76.6%) compared to the

**Table 2. Demographic Characteristics of Study Participants.**

| Description | Total (N = 102) | COVID-19 Positive Group (N = 72) | Non-COVID Group (N = 30) |
|---|---|---|---|
| **Mortality Rate** | 46.1% (N = 47) | 76.6% (N = 36) | 23.4% (N = 11) |
| **Gender Distribution** | | | |
| - Males | 49.0% (N = 50) | 51.4% (N = 38) | 40.0% (N = 12) |
| - Females | 51.0% (N = 52) | 47.2% (N = 34) | 60.0% (N = 18) |
| **Gender-based Mortality Rates** | | | |
| - Males | | 25.0% (N = 18) | 13.3% (N = 4) |
| - Females | | 25.0% (N = 18) | 23.3% (N = 7) |
| **Mean Age (years)** | 54 ± 14 | 57 ± 11 | 46 ± 19 |

**Table 3. Laboratory Values in COVID-19 Positive and Control Groups.**

| Laboratory Measure | COVID-19 Positive Group (Mean±SD) | Control Group (Mean±SD) | P-value |
|---|---|---|---|
| White Blood Cells (WBC) [×1000/µL] | 12.4±5.9 | 14.9±9.5 | 0.002** |
| Polymorphonuclear (PMN) [%] | 79±12 | 66±30 | 0.14** |
| Lymphocytes [%] | 13±10 | 15±14 | 0.94** |
| Hemoglobin (Hb) [g/dL] | 12.1±2.8 | 9.6±2.4 | <0.001* |
| Platelet Count [×1000/µL] | 242±111 | 126±116 | <0.001** |
| Erythrocyte Sedimentation Rate (ESR) [mm/h] | 72±36 | 79±34 | 0.38** |
| C-Reactive Protein (CRP) [mg/L] | 94±62 | 88±63 | 0.69** |
| Lactate Dehydrogenase (LDH) [U/L] | 836±428 | 835±525 | 0.88** |
| Aspartate Aminotransferase (AST) [U/L] | 38±34 | 40±49 | 0.34** |
| Alanine Aminotransferase (ALT) [U/L] | 46±37 | 34±25 | 0.25** |
| Alkaline Phosphatase (ALP) [U/L] | 287±263 | 297±360 | 0.41** |

*T-test.

** Mann-Whitney test.

non-COVID group (23.4%); however, this difference did not reach statistical significance ($P=0.21$). This suggests that while COVID-19 may exacerbate the condition, other factors also significantly influence the mortality outcomes in mucormycosis patients.

We assessed the laboratory values of deceased and surviving patients within each group to gain deeper insight into the factors contributing to mortality. Table 4 details these values, highlighting differences that might correlate with the observed mortality rates.

As presented in Table 4, significant findings in patients with a COVID-19 history include higher PMN percentages, CRP levels, and ALT levels in deceased patients. Conversely, survived patients had higher lymphocyte percentages and platelet counts. Interestingly, in patients without a COVID-19 history, significant differences were only in AST and ALT levels, with deceased patients showing elevated liver enzymes.

Based on Table 5, diabetes was significantly more prevalent among patients with a history of COVID-19. In contrast, the history of cancer was significantly more common in patients without a history of COVID-19, suggesting varied risk

**Table 4. Laboratory Values between Deceased and Surviving Patients in the COVID-19 History Positive Group and those without COVID-19 History.**

| Laboratory Measure | Patients with COVID-19 History | | | Patients without COVID-19 History | | |
|---|---|---|---|---|---|---|
| | Deceased Subgroup (Mean±SD) | Surviving Subgroup (Mean±SD) | P-value | Deceased Subgroup (Mean±SD) | Surviving Subgroup (Mean±SD) | P-value |
| White Blood Cells (WBC) [×1000/µL] | 13.1±6.3 | 11.7±5.4 | 0.29 | 14.7±22.7 | 6.4±6.9 | 0.5 |
| Polymorphonuclear (PMN) [%] | 83±9 | 75±13 | 0.011 | 60±39 | 68±26 | 0.85 |
| Lymphocytes [%] | 9±9 | 17±11 | 0.002 | 14±16 | 16±14 | 0.81 |
| Hemoglobin (Hb) [g/dL] | 11.5±3.0 | 12.7±2.5 | 0.07 | 9.4±2.3 | 9.8±2.4 | 0.54 |
| Platelet Count [×1000/µL] | 210±117 | 273±96 | 0.015 | 105±150 | 122±114 | 0.76 |
| C-Reactive Protein (CRP) [mg/L] | 117±68 | 71±47 | 0.004 | 97±69 | 80±59 | 0.55 |
| Lactate Dehydrogenase (LDH) [U/L] | 863±415 | 787±465 | 0.37 | 728±491 | 1051±597 | 0.3 |
| Aspartate Aminotransferase (AST) [U/L] | 42±30 | 32±38 | 0.018 | 62±65 | 21±11 | 0.045 |
| Alanine Aminotransferase (ALT) [U/L] | 49±36 | 42±40 | 0.11 | 50±30 | 21±9 | 0.035 |
| Alkaline Phosphatase (ALP) [U/L] | 261±150 | 323±372 | 0.41 | 391±513 | 212±92 | 0.36 |

**Table 5. Distribution of Pre-existing Medical Conditions.**

| Past Medical History | Total Patients | | | | With History of COVID-19 | | | Without the History of COVID-19 | | |
|---|---|---|---|---|---|---|---|---|---|---|
| | Total Patients | With History of COVID-19 | Without the History of COVID-19 | P-value | Deceased | Survived | P-value | Deceased | Survived | P-value |
| Diabetes History | 60 (58.8%) | 50 (83.3%) | 10 (16.7%) | 0.001 | 27 (54.0%) | 23 (46.0%) | 0.3 | 4 (40.0%) | 6 (60.0%) | 0.78 |
| Cancer History | 19 (18.6%) | 3 (15.8%) | 16 (84.2%) | < 0.001 | 3 (100%) | 0 (0.0%) | 0.23 | 5 (31.3%) | 11 (68.7%) | 0.51 |
| Immunosuppressive Drug History | 42 (41.2%) | 29 (69%) | 13 (31%) | 0.77 | 15 (51.7%) | 14 (48.3%) | 0.81 | 1 (7.7%) | 12 (92.3%) | 0.004 |
| Mortality | 47 (46%) | 36 (76.5%) | 11 (23.4%) | 0.21 | Not Applicable | | | | | |

profiles. As shown in Table 5, the use of immunosuppressive drugs correlated with a decreased risk of death in the control group, an observation that is not present in patients with a history of COVID-19.

As observed, diabetes and cancer were significant factors in the total population; therefore, their effect on laboratory tests and clinical symptoms cannot be undermined. To accurately assess the effect of previous COVID-19 infection, it is crucial to adjust the results for these significant factors. Therefore, GEE models were used, and the results were reported.

## Initial complaints and symptoms distribution

Table 6 outlines the initial clinical complaints of patients upon presentation, distinguishing between those with a history of COVID-19 and those in the control group without such a history. The distribution highlights significant differences in several symptoms, especially fever, where none of the COVID-19 history-positive patients reported this symptom, contrasting sharply with the control group, where all cases reported fever (P<0.001).

## Adjusted Impact of COVID-19 history on clinical symptoms and laboratory test

This section utilizes GEE to analyze the impact of previous COVID-19 exposure on a range of clinical symptoms and laboratory tests when adjusted with cancer, diabetes, and immunosuppressive history.

The GEE analysis in Table 7 has provided substantive insights into the differential impacts of COVID-19 positive history status and various pre-existing health conditions on a range of clinical symptoms. In this analysis, age and sex were also considered covariates. The detailed results pertaining to age and sex are presented in S2 File.

## Influence of COVID-19 history

The history of COVID-19 has demonstrated a statistically significant influence on several clinical parameters. Specifically, patients with a prior COVID-19 history exhibited a significant increase in platelet count (P=0.0311*), suggesting an ongoing alteration in hematological function post-infection. Additionally, this group showed a significantly reduced fever incidence (P<0.001**). A notable decrease was also observed in facial swelling (P=0.049*), indicating that post-COVID inflammatory processes might differ significantly from typical responses.

## Impact of past medical history of cancer

The history of cancer had significant impacts on various clinical outcomes. Hemoglobin levels and platelet counts experienced significant decreases (P<0.001**), indicating substantial reductions in these hematological parameters among cancer patients. Additionally, facial swelling significantly decreased (P<0.001**), reflecting physical changes related to the disease or its treatment. Facial parenthesis and ptosis also declined significantly (P<0.001**). Ophthalmoplegia, another significant decrease, further confirms the extensive effects of cancer (P<0.001**). Otalgia and dyspnea, too,

were significantly less common (*P<0.001\*\**), showing further systemic impacts. In addition to these decreases, CRP levels showed a highly significant increase (*P=0.0014\*\**), suggesting an inflammatory response related to cancer. The loss of consciousness (LOC) also had a highly significant positive association (*P<0.001\*\**), highlighting severe neurological impacts possibly related to cancer or its treatments.

### Influence of past medical history of diabetes

Patients with a history of diabetes were found to have significant increases in PMN (*P=0.011\**), highlighting an inflammatory response or altered immune function typical of chronic metabolic conditions. A highly significant rise in CRP (*P=0.0007\*\**) also emphasized an enhanced inflammatory state. Conversely, otalgia significantly decreased in this group (*P<0.001\*\**). In addition, otalgia and the loss of consciousness (LOC) demonstrated a significant increase (*P<0.001\*\**, and 0.036\*, respectively).

### Role of immunosuppressive treatment history

The history of immunosuppressive treatment significantly impacted several clinical biomarkers. There was a notable reduction in ESR (*P=0.019\**), and CRP levels also decreased (*P=0.045\**), reflecting significant changes in inflammatory markers among patients treated with immunosuppressives. Additionally, LDH levels showed a significant increase (*P=0.048\**), suggesting changes in cellular turnover or tissue state. Importantly, ptosis was also significantly more prevalent in this patient group, showing a positive association (*P=0.007\*\**), which adds to the clinical understanding of the broader impacts of immunosuppressive therapies on patients.

## Discussion

Our study addresses critical knowledge gaps in understanding the interplay between COVID-19 and mucormycosis, offering a distinctive contribution to the existing literature by examining how prior COVID-19 infection reshapes the clinical and

**Table 6. Distribution of Initial Complaints in COVID-19 Positive and Control Groups.**

| Initial Complaints | Total Patients (Percentage) | COVID-19 Positive Group (Percentage) | Control Group (Percentage) | P-value |
|---|---|---|---|---|
| Facial Paresis | 8 (7.8%) | 8 (100%) | 0 (0%) | 0.057 |
| Facial Paresthesia | 15 (14.7%) | 12 (80.0%) | 3 (20.0%) | 0.38 |
| Facial Pain | 15 (14.7%) | 14 (93.3%) | 1 (6.7%) | 0.03 |
| Facial Swelling | 15 (14.7%) | 10 (66.7%) | 5 (33.3%) | 0.71 |
| Periorbital Swelling | 19 (18.6%) | 17 (89.5%) | 2 (10.5%) | 0.04 |
| Sinus Pain | 5 (4.9%) | 4 (80%) | 1 (20%) | P>0.99 |
| Proptosis | 19 (18.6%) | 18 (94.7%) | 1 (5.3%) | 0.01 |
| Ptosis | 19 (18.6%) | 17 (89.5%) | 2 (10.5%) | 0.04 |
| Chemosis | 5 (4.9%) | 5 (100%) | 0 (0%) | 0.31 |
| Eye Pain | 9 (8.8%) | 7 (77.8%) | 2 (22.2%) | 0.62 |
| Ophthalmic Motility Disorder | 5 (4.9%) | 4 (80%) | 1 (20%) | P>0.99 |
| Blurred Vision | 30 (29.4%) | 23 (76.7%) | 7 (23.3%) | 0.38 |
| Ear Pain | 4 (3.9%) | 2 (50%) | 2 (50%) | 0.57 |
| Hearing Loss | 2 (2%) | 2 (100%) | 0 (0%) | P>0.99 |
| Nasal Discharge | 14 (13.7%) | 9 (64.3%) | 5 (35.7%) | 0.57 |
| Headache | 20 (19.6%) | 13 (65%) | 7 (35%) | 0.54 |
| Dyspnea | 5 (4.9%) | 4 (80%) | 1 (20%) | P>0.99 |
| Fever | 11 (10.8%) | 0 (0%) | 11 (100%) | P<0.001 |
| Loss of Consciousness | 10 (9.8%) | 9 (90%) | 1 (10%) | 0.15 |

**Table 7. Summary of GEE Results Evaluating the Impact of COVID-19 on Clinical and Laboratory Outcomes.**

| Variable | History of COVID-19 p-value | Past Medical History of Cancer p-value | Past Medical History of Diabetes p-value | Past Medical History of Immunosuppressive Treatment p-value |
|---|---|---|---|---|
| WBC | 0.2367 | 0.9587 | 0.1129 | 0.1767 |
| PMN | 0.789 | 0.115 | 0.011* (+) | 0.199 |
| Lym | 0.404 | 0.826 | 0.075 | 0.683 |
| HB | 0.569 | <0.001** (-) | 0.443 | 0.348 |
| PLT | 0.0311* (+) | <0.001** (-) | 0.4213 | 0.4762 |
| AST | 0.945 | 0.404 | 0.397 | 0.591 |
| ALT | 0.1318 | 0.6652 | 0.3813 | 0.4792 |
| ALP | 0.931 | 0.352 | 0.542 | 0.381 |
| ESR | 0.772 | 0.239 | 0.904 | 0.019* (-) |
| CRP | 0.3055 | 0.0014** (+) | 0.0007** (+) | 0.045* (-) |
| LDH | 0.713 | 0.956 | 0.644 | 0.048* (+) |
| Facial Parenthesis | 0.847 | <0.001** (-) | 0.398 | 0.144 |
| Facial Swelling | 0.049* (-) | <0.001** (-) | 0.033* (-) | 0.698 |
| Periorbital Swelling | 0.148 | 0.909 | 0.797 | 0.881 |
| Sinus Pain | 0.128 | 0.075 | 0.964 | 0.636 |
| Proptosis | 0.263 | 0.729 | 0.414 | 0.583 |
| Ptosis | 0.983 | <0.001** (-) | 0.563 | 0.007** (+) |
| Ophthalmic Pain | 0.326 | 0.381 | 0.711 | 0.902 |
| Ophthalmoplegia | 0.353 | <0.001** (-) | 0.681 | 0.721 |
| Blurred Vision | 0.64 | 0.71 | 0.23 | 0.33 |
| Otalgia | 0.397 | <0.001** (-) | <0.001** (+) | 0.737 |
| Nasal Discharge | 0.961 | 0.526 | 0.145 | 0.058 |
| Headache | 0.5 | 0.89 | 0.61 | 0.58 |
| Dyspnea | 0.52 | <0.001** (-) | 0.491 | 0.987 |
| Fever | <0.001** (-) | 0.52 | 0.9 | 0.36 |
| LOC | 0.281 | <0.001** (+) | 0.036* (+) | 0.166 |
| Mortality | 0.36 | 0.19 | 0.45 | 0.08 |

Note: Significance Indicators: * indicates $p < 0.05$ (statistically significant), ** indicates $p < 0.01$ (highly statistically significant). Directional Indicators: (+) indicates a positive association (increase in the outcome with the predictor), and (-) indicates a negative association (decrease in the outcome with the predictor). WBC (White Blood Cell count), PMN (Polymorphonuclear cells), Lym (Lymphocytes), HB (Hemoglobin), PLT (Platelets), AST (Aspartate Aminotransferase), ALT (Alanine Aminotransferase), ALP (Alkaline Phosphatase), ESR (Erythrocyte Sedimentation Rate), CRP (C-Reactive Protein), LDH (Lactate Dehydrogenase), LOC (Loss of Consciousness).

laboratory landscape of mucormycosis. Building on previous work that has recognized COVID-19-associated mucormycosis (CAM) as a complex entity marked by diagnostic challenges, our findings highlight that many observed differences in symptom presentation and laboratory parameters—such as platelet counts—cannot be fully attributed to COVID-19 infection alone, but rather emerge from a multifactorial context influenced by comorbidities like diabetes, cancer, and immunosuppression. By employing a multivariate modeling approach, we refine the understanding of how changes in laboratory profiles (e.g., shifts in platelet levels and inflammatory markers) and altered symptom patterns (e.g., fewer initial presentations with facial swelling and fever) may serve as more reliable diagnostic clues, ultimately informing clinical decision-making. This nuanced perspective not only advances the discourse on the overlap between COVID-19 and

mucormycosis, but also underscores the need for clinicians to adapt diagnostic strategies, interpret laboratory findings within a broader clinical context, and remain vigilant in identifying "masked" presentations of mucormycosis in patients with recent or concurrent COVID-19 infection.

Our study showed a significant correlation between the previous COVID-19 infection and the history of cancer or diabetes in these patients. Positive COVID-19 history was more prevalent among mucormycosis patients who had diabetes, and conversely, positive COVID-19 history was less common in mucormycosis patients who had cancer. The interplay between diabetes, COVID-19, and mucormycosis is complex, with conflicting results across various studies [23,24]; similarly, in our study, diabetes significantly changed the odds of developing multiple symptoms in various ways. Madhumitha M et al. found no positive correlation between diabetes as a comorbidity and mucormycosis; other research indicates that patients with diabetes mellitus have 4.9 to 6.7 times higher odds of developing post-COVID rhino-orbito-cerebral mucormycosis (ROCM) [25,26]. Additionally, a U.S. electronic health records study reported that SARS-CoV-2 infection increased the risk of new-onset diabetes mellitus by 65% compared to non-infected individuals [27]. These findings underscore the need for exploration of the interplay between diabetes and COVID-19, specifically in mucormycosis patients.

The relationship between COVID-19 and cancer as comorbidities in mucormycosis patients remains largely unexplored. While existing literature indicates that both COVID-19 and cancer are significantly correlated with the incidence of mucormycosis [28,29] Previous studies have primarily focused on the mortality and infection risk of COVID-19 in cancer patients and the potential risk of cancer development in post-COVID-19 patients [30–33]. Notably, there appears to be a gap in research explicitly examining the interplay of COVID-19 and cancer among individuals diagnosed with mucormycosis [34]. Our study addresses this gap, revealing an intriguing finding: a history of COVID-19 was significantly less common in mucormycosis patients who had cancer. We hypothesize that while these cancer patients inevitably developed mucormycosis, their underlying condition may have heightened their awareness of health risks, potentially leading to increased precautions against COVID-19 infection. This heightened vigilance which is explored by previous studies [35,36] could have contributed to a reduced rate of COVID-19 infection among cancer patients with mucormycosis. This observation suggests a complex relationship between these conditions, potentially involving behavioral factors and immune system interactions. It underscores the need for further investigation into the mechanisms underlying the interplay of COVID-19, cancer, and mucormycosis, as well as the role of patient behavior in modulating infection risks in immunocompromised populations.

The present study found a significant association between platelet counts and a history of COVID-19 status in mucormycosis patients, interestingly revealing that patient with a history of COVID-19 showing higher platelet levels. This finding is notable as it directly contradicts previous research by Shahcheraghi et al., which reported no significant difference in platelet levels between COVID-19 patients with mucormycosis based on various factors, including age and gender [37]. Our results also diverge from some studies, which observed decreased platelet counts in fungal septic patients [38]. Similarly, Mojtahedi et al. reported lower mean platelet counts in COVID-19 patients with rhinosinusitis mucormycosis [39]. Indeed, a review of the literature indicates that the typical expectation in COVID-19 and related fungal infections is either no change or a decrease in platelet counts. This divergence from previous research is likely attributable to our use of multivariate analysis, instead of the univariate models commonly employed in earlier studies. We discovered that some of the effects previously attributed solely to COVID-19 might actually be influenced by other risk factors and patients' medical histories. it is important to consider that altered platelet function, rather than just count, may be a critical factor in the pathogenesis of CAM. Studies have shown that COVID-19 patients exhibit increased platelet activation and altered gene expression, which can contribute to thrombotic complications [40]. The phenomenon is likely driven by the systemic inflammation inherent in both COVID-19 and mucormycosis, where the inflammatory milieu, characterized by elevated CRP and D-dimer in COVID-19, exacerbates platelet production [41–43].Clinically, these findings suggest that while thrombocytopenia may be a marker of severe COVID-19 [44,45], higher or normal platelet counts, particularly in the context of elevated inflammatory markers like CRP and D-dimer [41,42], in a patient with suspected mucormycosis and

a history of COVID-19 should not rule out the diagnosis. Instead, clinicians should consider the broader context of the patient's medical history and other laboratory findings, such as inflammatory markers, and recognize thrombocytosis as a potential indicator of CAM in the post-COVID-19 setting. This hypercoagulable state, combined with the vascular invasion characteristic of mucormycosis, could exacerbate tissue damage and contribute to the severity of the infection. Our multivariate analysis, which revealed an inverse correlation between cancer history and platelet counts, further highlights the complexity of interpreting platelet levels in the context of multiple comorbidities

Patients' age emerged as a significant predictor of mortality, with each additional year increasing the likelihood of death by approximately 4.83%. Conversely, a history of COVID-19 did not demonstrate a statistically significant association with higher mortality rates in the mucormycosis patient population when adjusted with diabetes, cancer and immunosuppression history.

The results of our study reveal intriguing patterns in the presentation of facial swelling among mucormycosis patients with and without a history of COVID-19. Notably, our analysis shows that patients with a history of COVID-19 had significantly lower odds of presenting with facial swelling. This finding aligns with emerging evidence suggesting that COVID-19 may alter the typical progression of mucormycosis, potentially leading to less pronounced facial swelling in the early stages [46]. The known cause of facial swelling is believed to be the invasion of facial tissues and blood vessels by the fungus, which leads to inflammation and edema. The underlying mechanism for typical progression alteration may be related to the immunomodulatory effects of SARS-CoV-2 infection. COVID-19 has been shown to induce lymphopenia, suppress T-lymphocyte function, and cause broader immune dysfunction [47], which could potentially delay or mask the inflammatory response characteristic of mucormycosis. Additionally, our results indicate that cancer and diabetic patients had significantly reduced odds of developing facial swelling, which may be attributed to their immunocompromised status and altered inflammatory response, as seen in COVID-19 patients. We agree with previous studies that the rapid progression of mucormycosis in COVID-19 patients may lead to detection through other symptoms before facial swelling becomes apparent, potentially masking this initial presentation [48]. While our proposed explanations for these patterns are plausible, it's important to recognize that we could not definitively confirm the precise underlying mechanism of altered facial swelling. Furthermore, due to the retrospective nature of our study and reliance on patient records where facial swelling was not consistently documented as a standardized symptom, the possibility of misclassification cannot be excluded, particularly in our cohort. These findings underscore the importance of maintaining a high index of suspicion for mucormycosis in post-COVID patients, even in the absence of typical facial swelling.

Our findings indicate a markedly reduced incidence of reported fever among mucormycosis patients with a history of COVID-19. While prior studies have not explicitly reported reduced fever in COVID-19-associated mucormycosis (CAM), they corroborate the diverse immunomodulatory effects of COVID-19, potentially manifesting as an afebrile presentation. Consistent with this, the cytokine storm characteristic of severe COVID-19 can suppress the febrile response to secondary infections, including mucormycosis [49]. Specifically, the dysregulation of key cytokines like IL-6, TNF-α, and IL-1β during COVID-19 may directly interfere with thermoregulation, contributing to a diminished febrile response in CAM patients with prior COVID-19 infection. This blunted fever response could potentially explain the significantly lower WBC levels observed in this patient group [49]. We also propose a multifactorial explanation for this. Firstly, while heightened surveillance in certain patient groups like cancer patients might lead to earlier detection before fever manifests. Secondly, we hypothesize that the presence of other debilitating symptoms could overshadow or modify the perception and reporting of fever. This phenomenon is not unique to COVID-19; for instance, cancer patients demonstrated significantly lower rates of reported symptoms, including otalgia, ophthalmoplegia, facial paresis, ptosis, and dyspnea, which highlights the need for comprehensive clinical assessment beyond traditional symptomatic indicators and emphasizing the importance of considering individual patient factors in diagnosis and treatment.

The remarkably high frequency of mucormycosis cases in our study—102 within a single year—is significantly greater than previously reported figures in Iran. For instance, a systematic review spanning 25 years (1990–2015) documented only 98 cases nationwide. in Khuzestan province, 20 biopsy-proven cases were reported over a decade from 2004 to 2014

[50], highlighting the rarity of mucormycosis during that period. Evidence of a rising trend existed even before the COVID-19 pandemic. A retrospective study covering 208 cases from 2008 to 2014 showed a rising trend, with cases increasing from 9.7% in 2008 to 23.7% in 2014 [51,52]. The sharp increase in our study aligns with recent findings during the COVID-19 pandemic. Fazeli et al. (2021) reported a significant surge in rhino-orbital mucormycosis cases among COVID-19 patients in Kermanshah, with risk factors including diabetes and immunosuppressive therapy such as corticosteroid use [53]. Additionally, Darazam et al. (2023) reported a 4.6-fold rise in cases during COVID-19 pandemic [54]. Our findings mirror this trend, underscoring the significant impact of the COVID-19 pandemic on the epidemiology of mucormycosis in Iran.

Several limitations warrant consideration in interpreting our findings. First, the study's retrospective design and reliance on self-reported history to determine prior COVID-19 infection status in the control group introduced potential selection bias, as we lacked definitive laboratory confirmation. Second, our dependence on existing patient documentation for symptom analysis, rather than standardized study-specific forms, may have affected data consistency. Third, the inherent limitations of COVID-19 PCR testing, including both false-positive and false-negative results, could have led to patient misclassification, potentially impacting the accuracy of our group assignments and subsequent analysis of COVID-19's relationship with mucormycosis. It should be noted that the positive group was considerably larger than the control group, which could further complicate our findings. Fourth, we were unable to determine the exact time interval between the onset of COVID-19 and the subsequent development of mucormycosis in our positive group. While our data confirmed that COVID-19 infection preceded mucormycosis diagnosis in all cases, the precise duration between these events remains unknown. Fifth, our dataset lacked detailed information on the severity of COVID-19 infection, an important factor that could influence the risk of developing mucormycosis. Finally, due to data anonymization protocols and the lack of a centralized vaccination database or electronic health record system in Iran, we were unable to assess the impact of SARS-CoV-2 vaccination status on the development of mucormycosis. Future prospective studies with systematic testing protocols, standardized data collection methods, and access to comprehensive patient data, including vaccination records, would help address these methodological constraints.

## Conclusion

In conclusion, this study reveals complex interactions between COVID-19 history, mucormycosis, and patient characteristics. Our findings challenge previous research by demonstrating significantly higher platelet counts in mucormycosis patients with a positive COVID-19 history. Notably, patients with prior COVID-19 infection showed lower odds of presenting with facial swelling and fever, potentially due to altered disease progression or immunomodulatory effects of SARS-CoV-2. These results underscore the importance of maintaining high clinical suspicion for mucormycosis in post-COVID patients, even without typical symptoms. Furthermore, our findings on altered laboratory parameters, including platelet counts and inflammatory markers, suggest the need for a revised diagnostic approach that integrates these findings with clinical assessment. This is particularly crucial given the potential for delayed or atypical presentation of CAM. Additionally, the study highlights the varied symptom presentation in different patient subgroups, such as reduced otalgia reporting in cancer patients and increased likelihood in diabetic patients. We have demonstrated that there are still gaps in the interplay of COVID-19, cancer, and diabetes among individuals diagnosed with mucormycosis, which is best explored by adopting multivariate approaches over univariate models to fully elucidate these intricate relationships.

## Supporting information

**S1 File. STROBE checklist.**
(DOCX)

**S2 File. GEE analysis results for covariates age and sex.**
(DOCX)

## Author contributions

**Conceptualization:** Sepideh Hejazi, Sahar Ravanshad, Reza Khoshbakht, Mehdi Bakhshaee.

**Data curation:** Ali Gholampour Kargar, Maryam Emadzadeh, Mona Kabiri, Reza Khoshbakht, Mohammad Hossein Ahmadi, Masoumeh Hosseinpoor, Mehdi Bakhshaee.

**Formal analysis:** Ali Gholampour Kargar, Arash Ziaee, Hamed Khosravi, Imtiaz Ahmed.

**Investigation:** Ali Gholampour Kargar.

**Methodology:** Ali Gholampour Kargar.

**Software:** Arash Ziaee, Hamed Khosravi, Imtiaz Ahmed.

**Supervision:** Sepideh Hejazi, Sahar Ravanshad, Maryam Emadzadeh, Mona Kabiri, Imtiaz Ahmed, Mehdi Bakhshaee.

**Validation:** Mehdi Bakhshaee.

**Writing – original draft:** Maryam Emadzadeh, Mona Kabiri, Reza Khoshbakht, Mohammad Hossein Ahmadi, Masoumeh Hosseinpoor, Hamed Khosravi.

**Writing – review & editing:** Sepideh Hejazi, Sahar Ravanshad, Arash Ziaee, Hamed Khosravi, Imtiaz Ahmed, Mehdi Bakhshaee.

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
