## [Decision Letter · Decision Letter 0]

10 Oct 2024

PONE-D-24-39871Impact of COVID-19 on Mucormycosis Presentation and Laboratory Values: A Comparative AnalysisPLOS ONE

Dear Dr. Bakhshaee,

Thank you for submitting your manuscript to PLOS ONE. After careful consideration, we feel that it has merit but does not fully meet PLOS ONE’s publication criteria as it currently stands. Therefore, we invite you to submit a revised version of the manuscript that addresses the points raised during the review process.

We look forward to receiving your revised manuscript.

Kind regards,

Hideo Kato

Academic Editor

PLOS ONE

Journal requirements: When submitting your revision, we need you to address these additional requirements. 1. Please ensure that your manuscript meets PLOS ONE's style requirements, including those for file naming. The PLOS ONE style templates can be found at https://journals.plos.org/plosone/s/file?id=wjVg/PLOSOne_formatting_sample_main_body.pdf and https://journals.plos.org/plosone/s/file?id=ba62/PLOSOne_formatting_sample_title_authors_affiliations.pdf 2. Please amend your list of authors on the manuscript to ensure that each author is linked to an affiliation. Authors’ affiliations should reflect the institution where the work was done (if authors moved subsequently, you can also list the new affiliation stating “current affiliation:….” as necessary). 3. We note that you have indicated that there are restrictions to data sharing for this study. For studies involving human research participant data or other sensitive data, we encourage authors to share de-identified or anonymized data. However, when data cannot be publicly shared for ethical reasons, we allow authors to make their data sets available upon request. For information on unacceptable data access restrictions, please see http://journals.plos.org/plosone/s/data-availability#loc-unacceptable-data-access-restrictions.  Before we proceed with your manuscript, please address the following prompts: a) If there are ethical or legal restrictions on sharing a de-identified data set, please explain them in detail (e.g., data contain potentially identifying or sensitive patient information, data are owned by a third-party organization, etc.) and who has imposed them (e.g., a Research Ethics Committee or Institutional Review Board, etc.). Please also provide contact information for a data access committee, ethics committee, or other institutional body to which data requests may be sent. b) If there are no restrictions, please upload the minimal anonymized data set necessary to replicate your study findings to a stable, public repository and provide us with the relevant URLs, DOIs, or accession numbers. Please see http://www.bmj.com/content/340/bmj.c181.long for guidelines on how to de-identify and prepare clinical data for publication. For a list of recommended repositories, please see https://journals.plos.org/plosone/s/recommended-repositories. You also have the option of uploading the data as Supporting Information files, but we would recommend depositing data directly to a data repository if possible. Please update your Data Availability statement in the submission form accordingly. 4. Your ethics statement should only appear in the Methods section of your manuscript. If your ethics statement is written in any section besides the Methods, please delete it from any other section.  5. Please include captions for your Supporting Information files at the end of your manuscript, and update any in-text citations to match accordingly. Please see our Supporting Information guidelines for more information: http://journals.plos.org/plosone/s/supporting-information. 

Reviewers' comments:

Reviewer's Responses to Questions

**Comments to the Author**

1. Is the manuscript technically sound, and do the data support the conclusions?

Reviewer #1: Yes

Reviewer #2: Yes

2. Has the statistical analysis been performed appropriately and rigorously? 

Reviewer #1: Yes

Reviewer #2: Yes

3. Have the authors made all data underlying the findings in their manuscript fully available?

Reviewer #1: Yes

Reviewer #2: Yes

4. Is the manuscript presented in an intelligible fashion and written in standard English?

Reviewer #1: Yes

Reviewer #2: Yes

5. Review Comments to the Author

Reviewer #1: The authors noted that this research aims to investigate potential differences in mucormycosis progression, initial symptom presentation, and laboratory value alterations in mucormycosis patients with COVID-19 history to enhance diagnostic accuracy.

Although it is an interesting initiative, several issues need to be resolved for publication.

These are described below.

Major comments

Page 7, Inclusion Criteria

The authors should describe what criteria they used to define mucormycosis with respect to culture and pathology, citing any already defined literature.

The frequency of mucomycosis was 102 cases per year at the two institutions more than past reports (10.1111/myc.12474). The appropriateness of the frequency in light of Iranian epidemiological data needs to be discussed along with the method of definition.

Similarly, the authors should describe in detail the symptoms defined as COVID-19. It should also be noted which manufacturer's equipment was used for PCR testing.

The authors should also describe how long after the onset of COVID-19 disease mucormycosis are included in the positive group; the positive group is considerably more numerous than the control group, and it would be desirable to compare the percentage of COVID-19-positive patients who developed fungal infections with the previous literature.

Page 11, Overview of Patient Clinical Characteristics

Authors should provide a breakdown of mycosis by species (candida, aspergills, etc.) and disease name for each group.

Page 15, Initial Complaints and Symptoms Distribution

The method section does not describe the extraction method, definition, etc. for initial clinical complaints. The authors should describe it in detail.

Results

It is not clear whether COVID-19 severity and length of hospitalization were included in the analysis without presentation of results. The authors need to clearly present or explain this point. The presence or absence of SARS-CoV2 vaccination and history of treatment for COVID-19 in eligible patients should also be presented.

Minor comments

Page 8, Data Collection and Variables

Table 2 shows the results of the extraction from the electronic records and should be shown in the results section, not the materials and methods section.

All tables

For every table, abbreviations should be annotated with their full names.

The number of cases in each category should be described (e.g.; n=xx).

Reviewer #2: The introduction offers a solid overview of the topic; however, it would be strengthened by a clearer statement of the research question and objectives, along with the inclusion of more recent references to emphasize the current state of research in this field.

The methodology section may also address potential limitations, such as the retrospective nature of the study or the possibility of selection bias.

There is no mention of how the sample size of 102 patients was determined. A lack of justification for the sample size can raise questions about the statistical power of the study.

There may be a need for a more nuanced interpretation of the results, particularly regarding the associations found between COVID-19 history and clinical outcomes. The implications of these findings should be discussed in the context of existing literature.

There may be a need for a more nuanced interpretation of the results, particularly regarding the associations found between COVID-19 history and clinical outcomes. The implications of these findings should be discussed in the context of existing literature.

The analysis should consider potential confounding variables that could influence the outcomes, such as the severity of COVID-19 or other underlying health conditions.

6. PLOS authors have the option to publish the peer review history of their article (what does this mean? ). If published, this will include your full peer review and any attached files.

**Do you want your identity to be public for this peer review?** For information about this choice, including consent withdrawal, please see our Privacy Policy .

Reviewer #1: No

Reviewer #2: No

---

## [Author Response · Author response to Decision Letter 0]

19 Nov 2024

Reviewer #1

Comment #1: The authors should describe what criteria they used to define mucormycosis with respect to culture and pathology, citing any already defined literature.

Response #1: We thank the reviewer for this important point. To clarify, our study relied solely on histopathological examination for the diagnosis of mucormycosis. The exact criteria for mucormycosis detection in histopathological examination was added with a citation to a recent paper about mucormycosis histopathology in COVID-19 patients. Please kindly refer to Inclusion Criteria Section.

Comment #2: The frequency of mucomycosis was 102 cases per year at the two institutions more than past reports (10.1111/myc.12474). The appropriateness of the frequency in light of Iranian epidemiological data needs to be discussed along with the method of definition.

Response #2: We appreciate the reviewer's comment regarding the epidemiological context of mucormycosis in Iran. To address this, we have added the following paragraph to the Discussion section:

 " The remarkably high frequency of mucormycosis cases in our study—102 within a single year—is significantly greater than previously reported figures in Iran. For instance, a systematic review spanning 25 years (1990-2015) documented only 98 cases nationwide. in Khuzestan province, 20 biopsy-proven cases were reported over a decade from 2004 to 2014 (43), highlighting the rarity of mucormycosis during that period. Evidence of a rising trend existed even before the COVID-19 pandemic. A retrospective study covering 208 cases from 2008 to 2014 showed a rising trend, with cases increasing from 9.7% in 2008 to 23.7% in 2014 (44, 45). The sharp increase in our study aligns with recent findings during the COVID-19 pandemic. Fazeli et al. (2021) reported a significant surge in rhino-orbital mucormycosis cases among COVID-19 patients in Kermanshah, with risk factors including diabetes and immunosuppressive therapy such as corticosteroid use (46). Additionally, Darazam et al. (2023) reported a 4.6-fold rise in cases during COVID-19 pandemic (47). Our findings mirror this trend, underscoring the significant impact of the COVID-19 pandemic on the epidemiology of mucormycosis in Iran.”

Comment #3: Similarly, the authors should describe in detail the symptoms defined as COVID-19.

Response #3: Based on your valuable feedback, we have incorporated a detailed list of clinical symptoms used to screen for COVID-19, which was derived from established literature (reference 16), including fever, cough with sputum production, smell and taste disturbances, fatigue, and shortness of breath. Additionally, we provided clear criteria for the control group selection.

 Added section: “Clinical symptoms suggestive of COVID-19 included fever, cough with sputum production, smell and taste disturbances, fatigue, and shortness of breath (16).”

Comment #4: It should also be noted which manufacturer's equipment was used for PCR testing.

Response #4: Thank you for your feedback. The manufacturing details are added in the following format:

 “For the molecular detection of COVID-19, the laboratory protocol employed the Rotor-Gene Q (Qiagen, Germany) PCR system coupled with the Magcore ® (RBC Bioscience Corp., Taiwan) extraction device to process and analyze patient samples.”

Comment #5: The authors should also describe how long after the onset of COVID-19 disease mucormycosis are included in the positive group; the positive group is considerably more numerous than the control group.

Response #5: We appreciate your insightful feedback regarding the temporal relationship between COVID-19 infection and subsequent mucormycosis development. In our retrospective cohort study, we utilized electronic health records to identify patients who developed mucormycosis following their COVID-19 infections. Through PCR testing, we confirmed that COVID-19 infection preceded the mucormycosis diagnosis in all cases. However, our dataset had limitations in determining the exact interval between initial COVID-19 onset and the subsequent development of mucormycosis. While we can definitively state that COVID-19 occurred first in every case, we were unable to establish precise time intervals between the two conditions for inclusion criteria in our COVID-19 positive group.

Comment #6: It would be desirable to compare the percentage of COVID-19-positive patients who developed fungal infections with the previous literature.

Response #6: Thanks for your comment. We would like to highlight that due to the cross-sectional nature of our study and limitations in our dataset, we were unable to assess the percentage of COVID-19 patients who developed mucormycosis. Our study focused on patients who were diagnosed with mucormycosis, and we collected data on their COVID-19 status at the time of mucormycosis diagnosis. However, we did not have access to the total number of COVID-19-positive patients in the broader population from which our mucormycosis cases were drawn. This means we could not calculate the incidence or proportion of mucormycosis among all COVID-19 patients.

Comment #7: Authors should provide a breakdown of mycosis by species (candida, aspergills, etc.) and disease name for each group.

Response #7: We once again thank for your feedback. Our study relied solely on histopathological examination for the diagnosis of mucormycosis therefore we don’t have details regarding breakdown of mycosis by species.

Comment #8: The method section does not describe the extraction method, definition, etc. for initial clinical complaints. The authors should describe it in detail.

Response #8: We have expanded the methods section to provide detailed information about our COVID-19 case definition and extraction methodology. Specifically, we added comprehensive criteria for identifying the exposure group, which included both current and historical COVID-19 infections. The definition encompasses both laboratory confirmation (positive PCR tests) and radiological evidence.

 Added paragraph: “Patients were included in the exposure group if they had either a current or historical COVID-19 infection. A current COVID-19 infection was defined by a positive COVID-19 PCR test or a chest X-ray highly suggestive of COVID-19. Additionally, individuals with a history of a positive COVID-19 PCR test at any point during the COVID-19 pandemic or chest X-ray findings indicative of a previous COVID-19 infection were also included in the exposure group. Chest X-ray findings suggestive of COVID-19 included primary features of atypical or organizing pneumonia, characterized by patchy or diffuse airspace opacities, whether consolidation or ground-glass opacity (14, 15). Patients without a positive COVID-19 PCR test and absence of these clinical symptoms were entered into the control group.”

Also, we acknowledge the inherent limitations in retrospective chart review studies, particularly regarding the standardization of symptom documentation. In our study, clinical symptoms were extracted from hospitalization records that were completed by medical students and residents during routine clinical care. We recognize that without prospectively established criteria, there may be variability in how different clinicians defined and documented these symptoms.

 We have explicitly acknowledged this as a study limitation in our manuscript.

• “Several limitations warrant consideration in interpreting our findings. The study's retrospective design and reliance on self-reported history to determine prior COVID-19 infection status in the control group introduced potential selection bias, as we lacked definitive laboratory confirmation. A notable limitation was our dependence on medical records completed by medical students and residents during routine clinical care, without standardized criteria for symptom definition and documentation. This variability in symptom reporting may have affected data consistency and completeness. The inherent limitations of COVID-19 PCR testing, including both false positive and false negative results, could have led to patient misclassification, potentially impacting the accuracy of our group assignments and subsequent analysis of COVID-19's relationship with mucormycosis. Future prospective studies with systematic testing protocols and standardized symptom assessment criteria would help address these methodological constraints.”

Comment #9: It is not clear whether COVID-19 severity and length of hospitalization were included in the analysis without presentation of results. The authors need to clearly present or explain this point.

Response #9: Thank you for your feedback. We agree that factors such as COVID-19 severity and hospitalization duration are crucial in understanding patient outcomes. Unfortunately, our dataset did not include detailed information on the severity of COVID-19 infection or the length of hospitalization for COVID-19. These variables were mistakenly included in Table 1 of the Methods section. We have corrected this error by removing these variables from the table and have revised the Methods section to accurately reflect the variables included in our analysis.

Comment #10: The presence or absence of SARS-CoV2 vaccination and history of treatment for COVID-19 in eligible patients should also be presented.

Response #10: We thank the respected reviewer for the valuable comment. Regarding vaccination status, due to our data anonymization protocol, patient identifiers (including national IDs) were removed from the dataset to protect patient privacy. Without these identifiers, we cannot retrospectively access vaccination records. Furthermore, there is no centralized vaccination database in Iran that would allow us to verify this information without patient identifiers. Also, concerning previous COVID-19 treatment history, Iran does not currently maintain a centralized electronic health record system, and patient treatment records are maintained separately by individual hospitals. Without patient identifiers, we cannot track treatments received at other healthcare facilities. The initial data collection did not include this historical information, and we cannot retroactively obtain it while maintaining patient anonymity.

Comment #11: Table 2 shows the results of the extraction from the electronic records and should be shown in the results section, not the materials and methods section.

Response #11: Thank you for your insightful comment, the table was moved and corrected based on your feedback.

Comment #12: For every table, abbreviations should be annotated with their full names.

The number of cases in each category should be described (e.g., n=xx).

Response #12: We appreciate the attention to details; the corrections were made based on your comment.

Reviewer #2

Comment #1: The introduction offers a solid overview of the topic; however, it would be strengthened by a clearer statement of the research question and objectives, along with the inclusion of more recent references to emphasize the current state of research in this field.

Response #1: Thank you for your insightful feedback. We would like to mention that the introduction’s last paragraph was rewritten in the following manner, and multiple citations were added regarding the current state of research.

 “This study aims to investigate the differences in progression and initial symptom presentation of mucormycosis among patients with and without COVID-19. Additionally, we will examine alterations in laboratory values in mucormycosis patients co-infected with COVID-19 compared to those without the COVID-19 virus. By addressing these objectives, we hope to contribute valuable insights into the clinical management of mucormycosis in the context of the COVID-19 virus, ultimately aiding in the development of more effective diagnostic and therapeutic approaches.”

Comment #2: The methodology section may also address potential limitations, such as the retrospective nature of the study or the possibility of selection bias.

Response #2: Thank you for your valuable comment; The limitation section has been updated as you suggested.

 “Several limitations warrant consideration in interpreting our findings. The study's retrospective design and reliance on self-reported history to determine prior COVID-19 infection status in the control group introduced potential selection bias, as we lacked definitive laboratory confirmation. A notable limitation was our dependence on medical records completed by medical students and residents during routine clinical care, without standardized criteria for symptom definition and documentation. This variability in symptom reporting may have affected data consistency and completeness. The inherent limitations of COVID-19 PCR testing, including both false positive and false negative results, could have led to patient misclassification, potentially impacting the accuracy of our group assignments and subsequent analysis of COVID-19's relationship with mucormycosis. Future prospective studies with systematic testing protocols and standardized symptom assessment criteria would help address these methodological constraints.”

Also, we would like to clarify that the sample of 102 individuals in this study represents the entirety of the available data rather than a subset selected for specific characteristics. This approach minimizes selection bias, as no intentional filtering or sampling was applied based on age, sex, or other demographic factors.

Figure 1: Age Distribution by Sex: a) Boxplot Analysis, b)Density Plot

Based on the age and sex distribution visualized in the plots, we conclude that there is no substantial bias in terms of these demographic factors:

Boxplot of Age by Sex:

The boxplot shows that both sex groups have comparable age distributions, with similar ranges and median ages. Although the age range for Sex 1 is slightly broader, the median ages for both groups are close, and their interquartile ranges (IQR) overlap significantly.

This overlap in IQRs and the lack of substantial differences in central tendencies indicate that the age distribution is balanced across sex groups, with no signs of preferential selection for particular ages within either group.

Density Plot of Age by Sex:

The density plot reveals slight differences in the shapes of age distributions for the two sex groups. Sex 0 has a single peak around middle age, while Sex 1 displays a broader distribution with multiple subtle peaks, suggesting greater variation in age within this group.

Despite these shape differences, the curves show considerable overlap, indicating that both sex groups include individuals across a wide range of ages. This overlap supports the notion that both sexes are well-represented and that any observed differences are due to natural variation rather than selection bias.

In summary, these analyses confirm that the age and sex distributions in our sample are representative of the available data, without any notable selection bias. Both sexes exhibit diverse age ranges, allowing for a balanced and unbiased analysis of age and sex effects in this study. However, we have also acknowledged in the limitations section that a larger sample could further enhance the ability to detect smaller effect sizes and refine estimates for non-significant predictors.

Comment #3: There is no mention of how the sample size of 102 patients was determined. A lack of justification for the sample size can raise questions about the statistical power of the study.

Table 1: Effect Sizes and 95% Confidence Intervals for Predictors Across Continuous Outcomes

Outcome Predictor Effect_Size CI_Lower CI_Upper

Lab_WBC

(Intercept) 11358.8 4199 22426.9

Post_covid1 2292.2 -1534 6271.7

Age -23.3 -162 84.7

Sex1 -997.8 -5736 2552.1

PMH_Cancer1 315.9 -7547 10712.1

PMH_DM1 2774.7 -1024 6297

r.Immunosuppressive1 -3160.9 -8276 1169.9

Lab_PMN

(Intercept) 69.9728 57.764 80.953

Post_covid1 0.6612 -6.568 7.546

Age -0.0197 -0.215 0.197

Sex1 6.9255 1.065 13.22

PMH_Cancer1 -10.4603 -26.365 1.831

PMH_DM1 7.3731 2.065 12.996

r.Immunosuppressive1 4.2642 -2.803 11.865

Lab_Lym

(Intercept) 19.5751 1

---

## [Decision Letter · Decision Letter 1]

10 Dec 2024

PONE-D-24-39871R1Impact of COVID-19 on Mucormycosis Presentation and Laboratory Values: A Comparative AnalysisPLOS ONE

Dear Dr. Bakhshaee,

Thank you for submitting your manuscript to PLOS ONE. After careful consideration, we feel that it has merit but does not fully meet PLOS ONE’s publication criteria as it currently stands. Therefore, we invite you to submit a revised version of the manuscript that addresses the points raised during the review process.

We look forward to receiving your revised manuscript.

Kind regards,

Hideo Kato

Academic Editor

PLOS ONE

Journal Requirements:

Reviewers' comments:

Reviewer's Responses to Questions

**Comments to the Author**

1. If the authors have adequately addressed your comments raised in a previous round of review and you feel that this manuscript is now acceptable for publication, you may indicate that here to bypass the “Comments to the Author” section, enter your conflict of interest statement in the “Confidential to Editor” section, and submit your "Accept" recommendation.

Reviewer #1: All comments have been addressed

Reviewer #2: All comments have been addressed

2. Is the manuscript technically sound, and do the data support the conclusions?

Reviewer #1: (No Response)

Reviewer #2: Yes

3. Has the statistical analysis been performed appropriately and rigorously? 

Reviewer #1: (No Response)

Reviewer #2: Yes

4. Have the authors made all data underlying the findings in their manuscript fully available?

Reviewer #1: (No Response)

Reviewer #2: Yes

5. Is the manuscript presented in an intelligible fashion and written in standard English?

Reviewer #1: (No Response)

Reviewer #2: Yes

6. Review Comments to the Author

Reviewer #1: The authors have generally responded appropriately to the points raised. Some regarding comments have been added and should be reviewed.

Comment #5: The authors should also describe how long after the onset of COVID-19 disease mucormycosis are included in the positive group; the positive group is considerably more numerous than the control group.

Response #5: We appreciate your insightful feedback regarding the temporal relationship between COVID-19 infection and subsequent mucormycosis development. In our retrospective cohort study, we utilized electronic health records to identify patients who developed mucormycosis following their COVID-19 infections. Through PCR testing, we confirmed that COVID-19 infection preceded the mucormycosis diagnosis in all cases. However, our dataset had limitations in determining the exact interval between initial COVID-19 onset and the subsequent development of mucormycosis. While we can definitively state that COVID-19 occurred first in every case, we were unable to establish precise time intervals between the two conditions for inclusion criteria in our COVID-19 positive group.

Regarding Comment #5

The authors should describe limitation that they were unable to research how long after the onset of COVID-19 disease mucormycosis are included in the positive group in limitation section.

Comment #9: It is not clear whether COVID-19 severity and length of hospitalization were included in the analysis without presentation of results. The authors need to clearly present or explain this point.

Response #9: Thank you for your feedback. We agree that factors such as COVID-19 severity and hospitalization duration are crucial in understanding patient outcomes. Unfortunately, our dataset did not include detailed information on the severity of COVID-19 infection or the length of hospitalization for COVID-19. These variables were mistakenly included in Table 1 of the Methods section. We have corrected this error by removing these variables from the table and have revised the Methods section to accurately reflect the variables included in our analysis.　

Regarding Comment #9

The authors should mention in the limitation that severity in particular is not investigated as it is an important factor.

Comment #10: The presence or absence of SARS-CoV2 vaccination and history of treatment for COVID-19 in eligible patients should also be presented.

Response #10: We thank the respected reviewer for the valuable comment. Regarding vaccination status, due to our data anonymization protocol, patient identifiers (including national IDs) were removed from the dataset to protect patient privacy. Without these identifiers, we cannot retrospectively access vaccination records. Furthermore, there is no centralized vaccination database in Iran that would allow us to verify this information without patient identifiers. Also, concerning previous COVID-19 treatment history, Iran does not currently maintain a centralized electronic health record system, and patient treatment records are maintained separately by individual hospitals. Without patient identifiers, we cannot track treatments received at other healthcare facilities. The initial data collection did not include this historical information, and we cannot retroactively obtain it while maintaining patient anonymity.

Regarding Comment #10

The authors should mention in the limitation that they have not been able to investigate vaccine status

Reviewer #2: Describe the study's distinctive contribution in relation to the body of previous literature, particularly in light of the overlap between COVID-19 and mucormycosis.

bolster the relationship between the possible diagnostic implications of changed laboratory findings and their clinical significance.

Include justification on GEE study used while statistical analysis

7. PLOS authors have the option to publish the peer review history of their article (what does this mean? ). If published, this will include your full peer review and any attached files.

**Do you want your identity to be public for this peer review?** For information about this choice, including consent withdrawal, please see our Privacy Policy .

Reviewer #1: No

Reviewer #2: No

---

## [Author Response · Author response to Decision Letter 1]

17 Dec 2024

Response to Reviewer Comments

We thank both reviewers for their insightful comments, which have significantly improved the manuscript. We have carefully addressed each point and made revisions accordingly.

Response to Reviewer #1

Thank you for your thorough review and helpful feedback. As suggested, we have thoroughly rewritten the Limitations section to more comprehensively convey the limitations of our study. This includes explicitly stating that:

• We were unable to determine the precise interval between COVID-19 onset and the development of mucormycosis due to data limitations.

• We were unable to investigate COVID-19 severity as a variable, as this information was not available in our dataset.

• We were unable to investigate vaccination status and past COVID-19 treatment due to the constraints of data anonymization and lack of a centralized database.

Response to Reviewer #2

We appreciate your feedback and have made the following changes to address your concerns:

• Statistical Analysis Section: We have updated the statistical analysis section to clearly state our intent for using Generalized Estimating Equations (GEE) for modeling correlated data within our study. We have also included a reference to support the use of GEE in our analysis approach.

• Introduction Closing Paragraph: We have revised the closing paragraph of the introduction to better highlight the specific goals of our study in light of the existing body of literature, focusing on the distinctive contributions of our research.

• Discussion Opening Paragraph: We have updated the opening section of the discussion to rephrase the significance of our findings and to emphasize the novel contribution our results make to the existing understanding of COVID-19 associated mucormycosis.

• Clinical Implications: We have included discussion that more clearly and directly points out the clinical implications of our findings, particularly in regards to the specific laboratory findings we report and their diagnostic value in the context of COVID-19 associated mucormycosis. We have also included 3 references to support the clinical significance.

• Conclusion Section: We have updated the conclusion section to reflect the changes and convey the revised goals of our study.

---

## [Decision Letter · Decision Letter 2]

30 Jan 2025

PONE-D-24-39871R2Impact of COVID-19 on Mucormycosis Presentation and Laboratory Values: A Comparative AnalysisPLOS ONE

Dear Dr. Bakhshaee,

Thank you for submitting your manuscript to PLOS ONE. After careful consideration, we feel that it has merit but does not fully meet PLOS ONE’s publication criteria as it currently stands. Therefore, we invite you to submit a revised version of the manuscript that addresses the points raised during the review process.

We look forward to receiving your revised manuscript.

Kind regards,

Hideo Kato

Academic Editor

PLOS ONE

Journal Requirements:

Reviewers' comments:

Reviewer's Responses to Questions

**Comments to the Author**

1. If the authors have adequately addressed your comments raised in a previous round of review and you feel that this manuscript is now acceptable for publication, you may indicate that here to bypass the “Comments to the Author” section, enter your conflict of interest statement in the “Confidential to Editor” section, and submit your "Accept" recommendation.

Reviewer #1: All comments have been addressed

Reviewer #2: All comments have been addressed

2. Is the manuscript technically sound, and do the data support the conclusions?

Reviewer #1: Yes

Reviewer #2: Partly

3. Has the statistical analysis been performed appropriately and rigorously? 

Reviewer #1: Yes

Reviewer #2: Yes

4. Have the authors made all data underlying the findings in their manuscript fully available?

Reviewer #1: Yes

Reviewer #2: Yes

5. Is the manuscript presented in an intelligible fashion and written in standard English?

Reviewer #1: Yes

Reviewer #2: (No Response)

6. Review Comments to the Author

Reviewer #1: The authors addressed the comments appropriately.

Reviewer #2: Explain why COVID-19 would reduce facial swelling in mucormycosis patients, or consider alternative explanations such as misclassification bias.

Expand the discussion section to compare results with prior studies, particularly regarding thrombocytosis, WBC count changes, and fever absence.

Provide references that support higher platelet counts in COVID-19 patients, or acknowledge that this finding contradicts prior studies.

Clarify whether verbal consent was formally recorded and approved by the ethics committee.

7. PLOS authors have the option to publish the peer review history of their article (what does this mean? ). If published, this will include your full peer review and any attached files.

**Do you want your identity to be public for this peer review?** For information about this choice, including consent withdrawal, please see our Privacy Policy .

Reviewer #1: No

Reviewer #2: No

---

## [Author Response · Author response to Decision Letter 2]

30 Jan 2025

Thank you for your insightful and constructive comments on our manuscript. We appreciate your thorough review and have revised the manuscript to address each of your points. Below is a point-by-point response outlining the changes we have made (changes are highlighted with green in the marked manuscript):

Point 1: Facial Swelling and COVID-19 Link

We acknowledge your concern regarding the seemingly counterintuitive notion of COVID-19 reducing facial swelling in mucormycosis patients. We agree that this aspect requires careful consideration. As you suggested, we have clarified in the discussion section that our explanations regarding facial swelling are not definitive and that misclassification bias cannot be ruled out due to the retrospective nature of our study and potential inconsistencies in symptom documentation. We have incorporated the following text into the discussion to explicitly address this point:

"[While our proposed explanations for these patterns are plausible, it's important to recognize that we could not definitively confirm the precise underlying mechanism of altered facial swelling. Furthermore, due to the retrospective nature of our study and reliance on patient records where facial swelling was not consistently documented as a standardized symptom, the possibility of misclassification cannot be excluded, particularly in our cohort]"

Point 2: Comparison with Prior Studies (Thrombocytosis, WBC, Fever Absence)

We appreciate your suggestion to expand the discussion section to compare our results with prior studies. We have now expanded this section to include comparisons with existing literature, particularly focusing on thrombocytosis, WBC count changes, and fever absence. We have incorporated four new references to support these expanded explanations.

Point 3: Platelet Counts in COVID-19 Patients

We have carefully considered your comment regarding the higher platelet counts observed in our COVID-19 associated mucormycosis (CAM) patients. We have clarified in the manuscript that while other studies may not have reported increased platelet counts in CAM patients specifically, our results indicate a significant increase in PLT in our cohort. We have reinforced our explanation for this finding, highlighting the role of systemic inflammation driven by both COVID-19 and mucormycosis. We have provided references to support the link between inflammation, elevated CRP and D-dimer in COVID-19, and increased platelet production. Furthermore, we have explicitly acknowledged that our finding regarding thrombocytosis in CAM patients may differ from studies that focus on thrombocytopenia in severe COVID-19 alone. The following text exemplifies this clarification in the discussion:

"[The phenomenon is likely driven by the systemic inflammation inherent in both COVID-19 and mucormycosis, where the inflammatory milieu, characterized by elevated CRP and D-dimer in COVID-19, exacerbates platelet production(41-43).Clinically, these findings suggest that while thrombocytopenia may be a marker of severe COVID-19(44, 45), higher or normal platelet counts, particularly in the context of elevated inflammatory markers like CRP and D-dimer(41, 42), in a patient with suspected mucormycosis and a history of COVID-19 should not rule out the diagnosis. Instead, clinicians should consider the broader context of the patient's medical history and other laboratory findings, such as inflammatory markers, and recognize thrombocytosis as a potential indicator of CAM in the post-COVID-19 setting.]"

Point 4: Verbal Consent and Ethics Approval

We have revised the manuscript to explicitly state that verbal consent was formally recorded and approved by the Ethics Committee. We have included the ethics code and detailed the process of verbal consent recording that was approved and implemented. The following text has been updated in the manuscript:

"[The study was approved with the ethics code IR.MUMS.MEDICAL.REC.1401.054. The Ethics Committee of Mashhad University of Medical Sciences waived the requirement for patients' written consent and specifically approved the use of verbal informed consent. Furthermore, the ethics committee approved the method for formally recording this verbal consent, and this approved procedure was implemented for all participants prior to their study involvement.]"

Thank you again for your time and valuable comments.

Sincerely,

---

## [Editor Report · Decision Letter 3]

14 Mar 2025

Impact of COVID-19 on Mucormycosis Presentation and Laboratory Values: A Comparative Analysis

PONE-D-24-39871R3

Dear Dr. Bakhshaee,

We’re pleased to inform you that your manuscript has been judged scientifically suitable for publication and will be formally accepted for publication once it meets all outstanding technical requirements.

Kind regards,

Hideo Kato

Academic Editor

PLOS ONE
---

## [Editor Report · Acceptance letter]

PONE-D-24-39871R3

PLOS ONE

Dear Dr. Bakhshaee,

I'm pleased to inform you that your manuscript has been deemed suitable for publication in PLOS ONE. Congratulations! Your manuscript is now being handed over to our production team.

Kind regards,

on behalf of

Dr. Hideo Kato

Academic Editor

PLOS ONE